# CD229 (Ly9) a Novel Biomarker for B-Cell Malignancies and Multiple Myeloma

**DOI:** 10.3390/cancers14092154

**Published:** 2022-04-26

**Authors:** Giovanna Roncador, Joan Puñet-Ortiz, Lorena Maestre, Luis Gerardo Rodríguez-Lobato, Scherezade Jiménez, Ana Isabel Reyes-García, Álvaro García-González, Juan F. García, Miguel Ángel Piris, Santiago Montes-Moreno, Manuel Rodríguez-Justo, Mari-Pau Mena, Carlos Fernández de Larrea, Pablo Engel

**Affiliations:** 1Monoclonal Antibodies Unit, Biotechnology Program, Spanish National Cancer Centre (CNIO), Centro de Investigación Médica en red Cancer (CIBERONC), 28029 Madrid, Spain; groncador@cnio.es (G.R.); lmaestre@cnio.es (L.M.); sjimenez@cnio.es (S.J.); aireyes@cnio.es (A.I.R.-G.); agarciag@cnio.es (Á.G.-G.); 2Immunology Unit, Department of Biomedical Sciences, Faculty of Medicine and Medical Sciences, University of Barcelona, Casanova 172, 08036 Barcelona, Spain; joanpunet91@gmail.com; 3Amyloidosis and Multiple Myeloma Unit, Department of Hematology, Hospital Clinic of Barcelona, 08036 Barcelona, Spain; lgrodriguez@clinic.cat (L.G.R.-L.); maripaumena@gmail.com (M.-P.M.); cfernan1@clinic.cat (C.F.d.L.); 4August Pi i Sunyer Biomedical Research Institute (IDIBAPS), 08036 Barcelona, Spain; 5Department of Pathology, MD Anderson Cancer Center Madrid, Centro de Investigación Médica en Red Cancer (CIBERONC), 28040 Madrid, Spain; jfgarcia@mdanderson.es; 6Department of Pathology, Fundación Jiménez Díaz, Centro de Investigación Médica en Red Cancer (CIBERONC), 28040 Madrid, Spain; miguel.piris@quironsalud.es; 7Translational Haematopathology Laboratory, Anatomic Pathology Department, Hospital Universitario Marqués de Valdecilla/IDIVAL, CIBERONC, 39008 Santander, Spain; santiago.montes@scsalud.es; 8Department of Research Pathology, Cancer Institute, University Collage London, London WC1E 6DD, UK; m.rodriguez-justo@ucl.ac.uk

**Keywords:** leukemia, lymphoma, myeloma, B cells, CD229, Ly9, soluble receptor

## Abstract

**Simple Summary:**

CD229 is a cell-surface molecule predominantly expressed on lymphocytes. Its expression in B-cell malignancies is poorly known. We tested the presence of this immunoreceptor on a large number of malignancies and normal tissue using a new monoclonal antibody and tissue microarrays. Our data show that CD229 expression is restricted to hematopoietic cells. It was strongly expressed in myeloma and marginal-zone lymphomas. Because of the high expression on multiple myeloma cells, we also analyze the presence of soluble CD229 in the sera of these patients. We showed that serum levels of soluble CD229 in myeloma patients, at the time of diagnosis, could be useful as a prognostic biomarker. Altogether, our results indicate that CD229 represents not only a useful disease biomarker but also an attractive therapeutic target.

**Abstract:**

CD229 (Ly9) homophilic receptor, which belongs to the SLAM family of cell-surface molecules, is predominantly expressed on B and T cells. It acts as a signaling molecule, regulating lymphocyte homoeostasis and activation. Studies of CD229 function indicate that this receptor functions as a regulator of the development of marginal-zone B cells and other innate-like T and B lymphocytes. The expression on leukemias and lymphomas remains poorly understood due to the lack of CD229 monoclonal antibodies (mAb) for immunohistochemistry application (IHC). In this study, we used a new mAb against the cytoplasmic region of CD229 to study the expression of CD229 on normal tissues and B-cell malignancies, including multiple myeloma (MM), using tissue microarrays. We showed CD229 to be restricted to hematopoietic cells. It was strongly expressed in all cases of MM and in most marginal-zone lymphomas (MZL). Moderate CD229 expression was also found in chronic lymphocyte leukemia (CLL), follicular (FL), classic mantle-cell (MCL) and diffuse large B-cell lymphoma. Given the high expression on myeloma cells, we also analyzed for the presence of soluble CD229 in the sera of these patients. Serum levels of soluble CD229 (sCD229) at the time of diagnosis in MM patients could be useful as a prognostic biomarker. In conclusion, our results indicate that CD229 represents not only a useful biomarker but also an attractive therapeutic target.

## 1. Introduction

CD229 is a type I cell-surface glycoprotein that belongs to the signaling lymphocytic activation molecule family (SLAMF) of cell-surface molecules. SLAMF receptors are a group of signaling molecules that are involved in the regulation of immune responses (reviewed in [1,2,3]). SLAMF receptors can either enhance or inhibit signals that regulate leukocyte activation, differentiation, and cytokine secretion. CD229, as with most of the SLAMF receptors, is activated by homophilic interactions, and therefore, acts as a self-ligand [4]. CD229 possesses four Ig-like domains, while the rest of SLAMF receptors contain two Ig-like domains. This molecule contains a long cytoplasmic tail with eight tyrosines, two of them embedded in immunoreceptor tyrosine switch motifs (ITSMs) [5,6]. They are considered switch motifs because of their ability to activate either kinases or phosphatases. The engagement of CD229 via its amino-terminal IgV-like domains induces the phosphorylation of the cytoplasmic tyrosines initiating unique signaling pathways. The primary ITSM binding partners are the SH2 domain-containing adaptor proteins, SLAM-associated protein (SAP) or EWS/FLI-activated transcript-2 (EAT-2), and tyrosine phosphatases such as SHP2 and SHIP1 [7,8,9]. The outcome of the signal delivered after receptor ligation depends on the interaction of these motifs with intracellular adaptor molecules or tyrosine phosphatases.

Studies from our group, using *Ly9*-deficient mice and agonistic monoclonal antibodies, indicated that this receptor acts as a negative regulator of the development and homeostasis of thymic innate-like CD8^+^, iNKT cells, and innate-like B cells such as marginal-zone B cells [10,11,12]. Moreover, aged *Ly9*-deficient mice spontaneously develop features of systemic autoimmunity indicating that the *Ly9* cell-surface receptor is involved in the maintenance of immune-cell tolerance [13].

Flow cytometry studies show that human CD229 is expressed on thymocytes, mature T and B cells, and a subset of NK cells [5,14]. In mice its highest expression levels are found on innate-like T lymphocytes such as iNKT and marginal-zone B cells [15]. Its expression is not lost during B-cell differentiation, and it is highly expressed on plasmablasts and plasma cells [16]. Consistently, several flow cytometry studies have found elevated levels of CD229 expression on malignant plasma cells in multiple myeloma (MM) cells [17,18,19]. In contrast, CD229 is not detected on bone marrow hematopoietic stem cells, bone marrow multipotent progenitors, monocytes, granulocytes, platelets, or red blood cells [5,20].

The absence of a mAb against CD229 that works on paraffin-embedded tissue and lack of reports describing its expression in reactive human tissues and B-cell lymphomas, prompted us to develop a specific mAb that could work on formalin-fixed paraffin-embedded (FFPE) tissues with which we could investigate the potential diagnostic value of CD229 as a marker for B-cell malignancies.

We also determined the levels of soluble CD229 (sCD229) as a potential serum biomarker for Multiple Myeloma (MM).

## 2. Materials and Methods

### 2.1. Production of an Anti-CD229 mAb for Immunohistology

A new anti-CD229 mAb (clone PIZCU426A) was produced by immunizing Wistar rats with the carboxyl terminal amino acids of CD229/Ly9 (NP_002339.2, residues 477–654) fused to a HIS-tag corresponding to the cytoplasmic domain that was produced in the BL21 strain of *Escherichia coli*. CD229-HIS was purified using a HIS-trap FF column (GE Healthcare, New York, NY, USA) connected to an ÄKTA-prime system (GE Healthcare, New York, NY, USA). Wistar rats (pathogen-free Wistar Han, female, 6 weeks old, 135 g weight, Charles River Laboratories, Lyon, France) were injected intraperitoneally (three times at 14-day intervals) with 100 μg of CD229-HIS fusion protein and Complete Freund’s adjuvant (BD, Cockeysville, MD, USA). Rats were housed within temperature, humidity, and room light ranges appropriate for the species. The cage used was a type IV 480 mm × 375 mm × 210 mm and animals had adequate bedding substrate and/or structures for resting and sleeping. Animal experiments were performed under the experimental protocol approved by the Institutional Committee for Care and Use of Animals from Consejería de Medio Ambiente y Ordenación del Territorio of the Comunidad de Madrid (Madrid, Spain) with reference project PROEX038/15. All efforts were made to minimize animal suffering.

A 150 μg last booster of the recombinant CD229-HIS protein was injected intraperitoneally and splenocytes were fused 3 days later (carbon dioxide was used for euthanasia). Hybridoma supernatants were screened by ELISA and using HEK293T cells transfected with pCI-neo-CD229 plasmid. The rat mAb that was raised against CD229 (clone PIZCU426A, IgG2a) was cloned by the limiting dilution technique.

### 2.2. Ly9 Gene Inactivation Using CRISPR/Cas9 Technology

The CRISPR/Cas9 knock-out generation was produced as previously described [21]. Briefly, sgRNAs were designed using the Benchling CRISPR sgRNA Design tool (http://www.bnchling.com, 13 January 2020). Specific sgRNAs were tested against *Ly9* gene (ENSG00000198846, exon 2) and a non-targeting (NT) guide was used as a control (sgLy9.1: TGGTGATGTCTAGTCGGCCC, sgLy9.2: GAACTCACCATAGACGAACA and sgNT: CCGCGCCGTTAGGGAACGAG). Ribonucleoprotein (RNP) complexes were formed at room temperature for at least 10 min by using in-vitro-synthesized guide sequences (IDT) and Cas9 protein (IDT).

U266 cells were electroporated using the Neon Transfection System (Thermo Fisher Scientific, Waltham, MA, USA). Before reaching confluence and 24 h prior to electroporation cells were passed in order to keep them in log phase; next day cells were resuspended into R solution and electroporated with 10 μL tips with 1 pulse for 20 ms (miliseconds) at 1650 v (volts). After electroporation, cells were maintained in 12-well plates with pre-warmed media and left to recover for 24 h. Single cells were sorted in a 96-well plate in order to establish single-cell CD229 KO clones and cultured in p96 plates.

### 2.3. Western Blot

WB analyses of CD229 protein were performed using total protein extracted from six cell lines lysed in a RIPA lysis buffer (Sigma-Aldrich, St. Louis, MO, USA) with protease inhibitors (Roche, Mannheim, Germany). The total lysates of each cell line were denatured by heating in Laemmli sample buffer, resolved on a 7.5% sodium dodecyl sulphate-polyacrylamide gel (SDS-PAGE) and transferred onto nitrocellulose membranes for 2 h. Blotting membranes were incubated overnight with blocking solution (5% milk in PBS) and immunoblotted for 1 h at room temperature with anti-CD229 PIZCU426A mAb (diluted 1:100), and anti-GAPDH monoclonal antibody (diluted 1:5000), followed by incubation with anti-mouse and rat AF680 (Invitrogen, Carlsbad, CA, USA) secondary antibody The blots were visualized using the Odyssey Image System (Omaha, NE, USA) in accordance with the supplier’s instructions. Information about commercial antibodies used is showed in Appendix A.

### 2.4. Human Tissues and Cell Lines

Labeling with the CD229 mAb (clone PIZCU426A) was performed across five types of reactive lymphoid tissues: lymph node, tonsil, bone marrow, thymus, and spleen, and 240 different lymphomas corresponding to 14 different subtypes and 20 multiple myelomas (Table 1). All the normal and tumor samples were retrospectively collected from the files of the participant institutions, in accordance with the technical and ethical procedures of the Spanish National Biobank Network, including anonymization processes and informed consent according to the Helsinki Declaration. Approval was obtained from the Clinical Research Ethical Committee (min no. 10/18 with code PIC075-18_FJD).

RPMI-8226, SKMM2, U266, AKATA, DAUDI, RAMOS, and RAJI cell lines used in the present study were obtained from the German Collection of Microorganisms and Cell Cultures (DSMZ, Braunschweigh, Germany). K562, SU-DHL-4, JURKAT, THP-1, REH, REC.1, YT, and CESS-EBV cell lines were purchased from the American Type Culture Collection (ATTC, Manassas, VA, USA).

HEK293T and KARPAS-620 cell lines were kindly provided by Miguel Angel Piris from Department of Pathology, Fundación Jiménez Díaz, Spain. Cell lines were authenticated by short tandem repeat (STR) profiling.

### 2.5. Immunohistochemistry

FFPE tissues from many of tumor samples were included in four tissue micro array (TMA) (one multi lymphoma, one with MALT, 1 one NMZL, and one with DLBCL lymphomas) blocks using a Tissue Arrayer Device (Beecher Instrument, Silver Spring, MD, USA). IHC analyses were performed on TMAs or full tissue sections. Antibody sources are described in Appendix A. The Bond Polymer Refine detection system and Bond RX automated stainer system (Leica Biosystems, Wetzlar, Germany) were used for the immunoenzymatic labeling of FFPE tissues. Images were captured with an Axiocam charge-coupled device (CCD) camera (Zeiss, Jena, Germany) and Axiovision software ZEN2.1 (Imaging Associates, Bicester, UK) and adjusted using Photoshop version 9.0 software (Adobe, San Jose, CA, USA).

### 2.6. Scoring CD229 Expression

CD229 protein expression was assessed by two independent observers (J.F.G. and M.R.J.) by immunohistochemistry on reactive as well as neoplastic human TMA and on complete sections. Since most tumors showed uniform immunohistochemical expression, we selected a score of 10% as the most informative cut off. Each case was scored semi-quantitatively, depending on the number of positive cells, as negative (0–10% positive tumor cells) or positive (10–100% positive tumor cells).

### 2.7. Immunofluorescence

After antigen retrieval (Tris-EDTA buffer), the slides were incubated for one hour at room temperature in a humid chamber with primary antibodies (anti-CD229 clone PIZCU426A, anti-PD1 clone NAT105C, anti-CD3 clone IR503, and anti-CD20 clone SP32 (Appendix A). Slides were then washed in PBS 0.5%-Tween20 (Sigma-Aldrich, Saint Louis, MO, USA) three times for 5 min each. The slides were incubated for 1 h with fluorochrome-conjugated antibodies (Alexa Fluor 555, Alexa Fluor 488, and Alexa Fluor 680, dilution 1:200, Invitrogen, Carlsbad, CA, USA) against the different species, diluted in PBS (Molecular Probes, Leiden, The Netherlands) in a humid chamber in the dark. Subsequently, slides were washed in PBS 0.5%-Tween20 (Sigma-Aldrich, Saint Louis, MO, USA) three times for 5 min each. Following washing, antifading medium with Dapi (Qbiogene, Illkirch, FR) was added. Slides were examined on a Leica TCS SP8 STED 3X (Leica, Wetzlar, Germany) confocal microscope equipped with a 63 ×/NA 1.4 oil immersion objective, a 405 laser, a white light laser, and multispectral HyDTM detectors. Fluorescence images were captured with LAS X Navigator software v3.5 (Leica, Wetzlar, Germany), and adjusted using Photoshop version 9.0 software (Adobe, San Jose, CA, USA).

### 2.8. Flow Cytometry

Cell lines were counted, washed, and resuspended in PBS, 20% inactivated rabbit serum, 6% FBS, and 0.09% NaN3. Then, 1 × 10^6^ cells were incubated with the anti-human CD229-PE (clone Ly9.1.25, BD, Jersey City, NJ, USA) on ice and protected from light for 45 min. After washing once, labelled cells were acquired on an LSRII Fortessa cytometer (BD). FlowJo vX.0.7 (Tree Star, Inc., Ashland, TN, USA) software were used for the analysis.

### 2.9. Myeloma Patients

The clinical records of 221 patients with a monoclonal gammopathy who had been followed at the Hospital Clínic of Barcelona were reviewed. A total of 144 patients had been diagnosed with MM (122 patients with newly diagnosed MM (NDMM), 39 patients achieving response after anti-myeloma therapy, and 18 patients with MM after disease relapse), while 23 had smoldering MM (SMM) and 54 a monoclonal gammopathy of undetermined significance (MGUS) (Table 2). The diagnoses were made according to the criteria of the International Myeloma Working Group [22]. The study was approved by the Ethics Committee of the Hospital Clínic of Barcelona and was in accordance with the Declaration of Helsinki. All patients signed informed consent in accordance with our institutional requirements.

### 2.10. Soluble CD229 ELISA

High binding plates (Corning, New York, NY, USA) were coated overnight with 3 µg/mL of the anti-human CD229 antibody (clone LY9.1.84) which recognizes the second immunoglobulin domain of CD229 (produced in our laboratory) diluted in PBS [4]. Wells were blocked with PBS 2%BSA and washed with PBS-Tween. Sera from patients was diluted 1/10 in PBS 2%BSA and undiluted cell culture supernatants were incubated for 1 h at room temperature. Human CD229-Fc protein (produced in our laboratory) was used as standard [4]. To detect captured sCD299 biotin-anti-humanCD229 (clone Ly9.1.25), which recognizes the first immunoglobulin domain of CD229 (produced in our laboratory), was added [4]. Streptavidin HRP (Roche) and TMB substrate (BD Bioscience) were used to develop the plates. Readings were carried out on Epoch plate reader at 450–570 nm. Limit of detection corresponded to 0.0423 ± 0.0019 O.D. and limit of quantification corresponded to 0.15 ng/mL of sCD229.

To test the production of sCD229 from different cell lines, we seeded 1 × 10^5^ cells per well into 96 flat-bottomed plates (JET-BIOFIL, Guangzhou, China). After 48 h, supernatants were collected and immediately used to detect sCD229 by ELISA. The experiment was carried out with triplicates. Sera were diluted 1/10 in PBS 2%BSA.

### 2.11. Statistical Analysis

Frequency and percentage were used for categorical variables while median and range were used for continuous variables. Differences between groups were compared using Fisher’s exact test or c2 test for categorical variables, as well as the *t*-test or Wilcoxon test for continuous variables. Spearman’s rho correlation coefficient was used to compare continuous variables. The starting point for time-to-event analysis was the date of diagnosis. Prognosis was determined by analyzing PFS and OS. The starting point for both was first line treatment initiation. The probability of PFS and OS was calculated with the Kaplan–Meier method and survival curves were compared using the log-rank test. The Cochran–Mantel–Haenszel statistic was performed to stratify data and adjust for confounding variables. Multivariate Cox regression models were performed using the backward stepwise method to identify factors independently associated with PFS and OS. The method of maximally selected rank statistics (maxstat and survmine packages, R software) was used to calculate the best sCD229 cut-off to predict PFS and OS, and a value of 5.0 ng/mL was obtained (Appendix A). P values were two-sided and *p* < 0.05 indicated statistical significance. All analyses were performed using R.3.6.1 (R Foundation for Statistical Computing, Vienna, Austria) and GraphPad Prism version 8.0.2.

## 3. Results

### 3.1. CD229 Tissue Expression is Restricted to Hematopoietic Cells

A new mAb (clone PIZCU426A, rat isotype IgG2a/k), was generated in order to test the expression of CD229 in formalin-fixed paraffin-embedded tissues (FFPE). It was raised against the cytoplasmic tail of CD229 protein. The specificity of PIZCU426A mAb for the endogenous CD229 protein was confirmed by Western blotting (WB) using cell extract of the HEK293T transfected with an unrelated protein (IL11RA) versus HEK293T-CD229 cells (Appendix A), U266 cell line before and after *LY9* gene inactivation using CRISPR-Cas9 technology (Appendix A) and positive and negative human cell lines and tonsil (Appendix A). WB with this mAb showed the two characteristic bands of CD229 ~120 kDa and ~100 kDa that corresponded to the full-length CD229 and a shorter spliced isoform [5].

In the tonsil and lymph node, CD229 was strongly expressed in mature B and T cells from the interfollicular area and in plasma cells present in the subepithelial area outside the follicles, as well as in the mantle zone (Figure 1A,B). Weaker CD229 expression was found in B and T cells of the germinal center (GC) light zone while the dark zone was negative (Figure 1A,B). In the thymus, CD229 was found mainly in mature T cells in the medulla while weaker expression was detected in the cortex (Figure 1C). In the spleen, the mantle and marginal zone were strongly positive, as well as some T cells within the white-pulp area (Figure 1D). These observations indicate a progressive increase in the expression of CD229 during T and B cell maturation, with its highest expression on plasma cells.

In the thymus, CD229 was found mainly in the medulla while in the spleen strong staining was observed in the mantle and marginal zone (image 40×).

We further analyzed the phenotype of the CD229-positive cells in the tonsillar GC cells using triple immunostaining. As shown in Figure 2, CD229-positive cells were mainly CD3+PD1+ cells that corresponded to T follicular helper cells (TFH). The majority of the CD20 cells surrounding the PD1+/CD229+ TFH did not express CD229.

In contrast, all the normal non-hematopoietic human samples tested (lung, brain, thyroid pancreas, small intestine, duodenum, colon, stomach, endometrium, uterus, testicle, prostate, bladder, kidney, skin and placenta) were negative except for the lymphoid component present in these tissues, such as in the gastrointestinal tract (Figure 3). These data confirm that CD229 expression is restricted to hematopoietic cells.

### 3.2. CD229 Expression on B Cell Linage Lymphomas and Myeloma

The immunostaining results from paraffin sections of 260 primary B-cell lymphomas are summarized in Table 1. CD229 was expressed in a large number of B-cell neoplasms (Figure 4) representing different differentiation stages. In particular, CD229 expression was lower in B-lymphoblastic lymphoma (L-BL) (28%) and a similar result was found in Burkitt lymphoma (BL) (20%). Higher and heterogeneous expression was found in chronic lymphocytic leukemia (CLL) (70%), classic mantle-cell lymphoma (MCL) (70%), follicular lymphoma (FL) (60%), and diffuse large B-cell lymphoma (DLBCL) (60%). When DLBCLs were classified as germinal center B-cell-like (GCB) and non-germinal center B-cell-like (non-GCB) subtype accordingly to the Hans algorithm [23], we found that CD229 expression was independent of DLBCL origin (GCB, 66%; non-GCB, 52%).

In B-cell lineage lymphomas from post-GC B cells, such as marginal-zone lymphomas (MZL) and myeloma, CD229 expression was very high. High expression of CD229 was found in nodal marginal-zone lymphoma (NMZL) (82%), mucosa-associated lymphoid tissue lymphoma (MALT) (86%), and splenic marginal-zone lymphoma (SMZL) (87%) (Table 1 and Figure 4). In myeloma, CD229 expression was very high and detected in 100% of the cases (Table 1 and Figure 5).

In contrast, CD229 was not found in tumor cells in Hodgkin lymphoma (HL), although it was highly abundant in the tumor microenvironment of nodular lymphocyte-predominant Hodgkin lymphoma (NLPHL) (Figure 4).

### 3.3. Soluble CD229 (sCD229) Is Secreted by B-Cell Lymphoma and Myeloma Cells Lines

We studied the expression levels of CD229 of several B-cell lymphoma and myeloma cell lines using flow cytometry. Consistent with the data shown above, the myeloma cell lines were the ones that presented the highest expression levels (Figure 6A). We tested the capacity of these cell lines to secrete sCD229, after 48 h of culturing them without any stimuli, using an ELISA. We observed a correlation of CD229 expression intensity on leukocyte cell lines and the levels of sCD229 present in their supernatants (Figure 6B).

### 3.4. Soluble CD229 as a Biomarker in MM

The identification of potential prognostic biomarkers is helpful in predicting relapse and survival in patients with cancer. Therefore, to figure out the possible relationship between sCD229 levels and outcomes in patients with MM and asymptomatic monoclonal gammopathies, we evaluated sCD229 levels in the serum of patients with monoclonal gammopathy of undetermined significance (MGUS), smoldering multiple myeloma (SMM), newly diagnosed multiple myeloma (NDMM) and MM during response and after relapse using sandwich ELISA. sCD229 levels were statistically significantly higher in patients with active MM as compared with those with MGUS or SMM (Figure 7A). The samples analyzed from patients in response after treatment showed a significant decrease in the levels of sCD229. An elevation of this protein was observed again in those patients who relapsed (Figure 7A). Among NDMM patients, serum levels were considerably increased in advanced vs. early stages (International Staging System (ISS) stage II and III vs. I) (Figure 7B).

We observed a moderate positive correlation between the amount of sCD229 with the percentage of bone marrow plasma cells (BMPC) (R = 0.486; *p* < 0.001) and serum β2-microglobuline (R = 0.419; *p* < 0.001) (Appendix A).

Later, we analyzed the clinical and laboratory differences of the 122 patients with NDMM according to the established cut-off of 5.0 ng/mL (Table 2). Ninety-nine patients (81%) had a sCD229 below 5.0 ng/mL. Patients with an sCD229 > 5.0 ng/mL were significantly older with a higher percentage of BMPC, as well as levels of β2-microglobuline and creatinine, with lower levels of hemoglobin, platelets and, albumin. The proportion of patients with stage III ISS score was higher among patients with a high sCD229 (82.6 vs. 31.6%, *p* < 0.001). No differences were seen regarding sex, immunological subtype, light chain isotype, calcium or lactate dehydrogenase levels, or the presence of lytic lesions. These data suggest that patients with higher levels of sCD229 have a higher tumor burden and more aggressive clinical features. In terms of predictive value (overall response and complete remission rate), there were no relationship with serum sCD229 levels (Table 2).

Regarding prognosis (impact on progression-free survival (PFS) and overall survival (OS)), median PFS was shorter after first line of treatment for NDMM patients with sCD229 levels > 5.0 ng/mL compared to those with levels ≤5.0 ng/mL (16.1 vs. 25.5 months; hazard ratio (HR) 1.97; *p* = 0.01) (Figure 8A). Furthermore, the 5-year OS was shorter among those with sCD229 levels >5.0 ng/mL compared with those ≤5.0 ng/mL (39.1 vs. 61.0%; HR 2.3; *p* = 0.01) (Figure 8B) However, when variables related to a worse prognosis in patients with newly diagnosed MM were included in the multivariate Cox regression models for PFS and OS, it was observed that the levels of hemoglobin, albumin and sCD229 lost their prognostic impact, but not age and β2-microglobulin (Appendix A). These results hint that serum levels of sCD229 at the time of diagnosis in patients with MM could be useful as a prognostic biomarker.

## 4. Discussion

This study provides a detailed description of the distribution of CD229 cell-surface protein in a wide variety of normal and malignant human tissues. The results were obtained using a novel anti-CD229 mAb (clone PIZCU426A), suitable for immunohistochemical staining of formalin-fixed paraffin-embedded tissue sections and WB analysis.

Novel finding in this paper show that CD229 expression was restricted to lympho-hematopoietic tissues, mainly on B cell linage cells with high expression levels on mature B cells. CD229 was also found in splenic marginal and mantle-zone B cells. Interestingly, marginal-zone B cell numbers are increased in CD229-deficient mice, indicating that CD229 plays a key role in the development of this cells type [12]. However, the highest expression levels were found on terminally differentiated plasma cells. These data agree with those reported with flow cytometry using blood cells, tonsil and spleen cells where the CD229 was found on mature B cells, with the highest expression levels on plasmablasts and plasma cells [15,16]. In contrast to these reports, we find a low CD229 expression on in germinal center B cells, especially in the dark zone.

Here, we also show the expression of CD229 on thymic T cells located in the medulla. Mature T cells were also stained in the interfollicular areas of tonsil and spleen. Interestingly, triple immunofluorescence staining also showed that the majority of T follicular helper CD3+/PD1+ cell present in the GC also expressed CD229. This is in concordance with the observation made with gene microarrays, which identified CD229 as a molecule preferentially expressed in T follicular helper cells [24].

We also investigated CD229 expression in a large number of B-cell lymphomas using tumor tissue microarrays. In agreement with the observed expression pattern on lymphoid tissues, CD229 expression in B cells lymphomas seems to progressively increase depending on the stage B cell maturation with its higher expression in mature B cells. Particularly, CD229 was highly expressed in marginal-zone lymphomas (SMZL (87%), MALT (86%) and NMZ (82%) lymphomas) while heterogeneous expression was found in CLL (70%), MCL (70%), FL (60%) and DLBCL (GCB type 66% and non-GCB type (52%). When DLBCLs were classified in GCB and non-GCB subtype accordingly to the Hans algorithm we found that CD229 was similarly expressed in the two subtypes. Low CD229 was found in BL. CD229 was not found in tumor cells in cHL, although was highly abundant in the tumor microenvironment of NLPHL. In agreement with is high expression on plasma cells, all MM cases were positive. This has been already reported by others studying myeloma cells in bone marrow [17,18,19]. Moreover, it has been suggested that CD229 could have a relevant biological role in the survival of myeloma cells [25], and that its expression is higher in myeloma-propagating cells, which are more quiescent, and more drug-resistant than the common malignant plasma cell [26].

Furthermore, this study showed that the measurement of sCD229 levels could provide important diagnostic and prognostic data for patients with MM. sCD229 levels correlated with the percentage of BMPC infiltration from NDMM. It was observed that in more initial or indolent stages of the monoclonal gammopathy spectrum (MGUS and SMM) there was a limited serum concentration, having potential diagnostic relevance in these patients. Further, those patients who achieved a response had a reduction in sCD229 serum concentration, while on relapse the expression increased again. The maximally selected rank statistics method showed that the cut-off point of 5 ng/mL would be the best threshold to detect prognostic differences. Elevated levels of sCD229 above cut-off were associated with more aggressive clinical features and higher tumor burden. We also examined the capability of sCD229 to predict PFS and OS, finding that those patients with sCD229 levels above 5 ng/mL at the time of diagnosis had a shorter PFS and OS. However, in the multivariate analysis, the prognostic impact of sCD229 was lost. This could be explained due to the relatively small number of cases who presented levels of sCD229 above 5 ng/mL. This is also reflected by observing that variables such as levels of hemoglobin and albumin also lose their negative prognostic impact. The results presented in this article are in concordance with those recently published by Ishibashi et al. [27]. In this case, cut-off value for detecting high-risk patients was 3.3 ng/mL. This variation could be due to the differences between the populations analyzed, ELISA technique, and the method used to determine the best cut-off point. Nevertheless, both results support the usefulness of sCD229 as a novel prognostic biomarker in MM.

Recently, the use of immunotherapy, such as chimeric antigen receptor (CAR) T-cell therapy against B-cell maturation antigen (BCMA) and bispecific antibodies in patients with MM has shown encouraging results [28,29,30,31,32]. The possible role of the soluble fraction of BCMA (sBCMA) as a possible prognostic biomarker has been elucidated [33], even after the use of anti-BCMA CAR T-cell therapy [34]. However, there is preclinical evidence suggesting that sBCMA may interfere with CAR T-cell activity [33]. Regarding CD229, there are already preclinical data that showed that CAR T-cells directed against this protein could be a promising strategy in MM [35] due to its central role in the pathophysiology of MM, its strong and homogeneous expression in myeloma cells and myeloma-propagating precursor cells, and their lack of expression in most normal tissues [25]. However, potential disadvantages of CD229 targeted therapies could be the development of “on-target off-tissue” toxicity due to its expression in NK cells and T cells and the possibility that sCD229 could hamper CAR T-cell function in patients with MM. In this sense, Radhakrishnan et al. [35] recently produced and tested a CAR T cell against CD229. They demonstrated that the CD229 CAR T cells were highly active in vitro and in vivo against MM plasma cells, memory B cells, and MM-propagating cells. They did not observe fratricide during CAR manufacture due to a downregulation of the CD229 protein in activated T cells. In contrast, T cells with high expression of CD229 but not low or negative expression were killed by the CD229 CAR T cells. This could lead to prolonged cellular immunosuppression in patients with myeloma after receiving a CAR T cell therapy directed against this target antigen. Therefore, therapeutic strategies will be required to facilitate immune reconstitution (stem cell transplantation) or limit the persistence of the CAR T cell by integrating suicide genes into the CAR construct.

In summary, CD229 mAb will provide a valuable research tool to support further investigations into the role of CD229 as marker of plasma cells in both routine clinical samples and pre-clinical test. Specific and reproducible detection of CD229+ plasma cells by flow cytometry and IHC, and sCD229 by ELISA could help to detect and to monitor patients with multiple myeloma. Moreover, the data about expression of CD229 in B-cell lymphomas will be useful to extend the potential target diseases that can benefit with treatments targeting CD229.

## 5. Conclusions

Our results show that CD229 represents a new biomarker of B-cell malignancies, especially in MM. We also show that higher levels of sCD229 is associated with more aggressive clinical features and higher tumor burden, and it could also be used as a prognostic biomarker for patients with MM. Moreover, our data also support the idea of CD229 as an excellent therapeutic target for the treatment for MM and other B-cell malignancies.

## Figures and Tables

**Figure 1 cancers-14-02154-f001:**
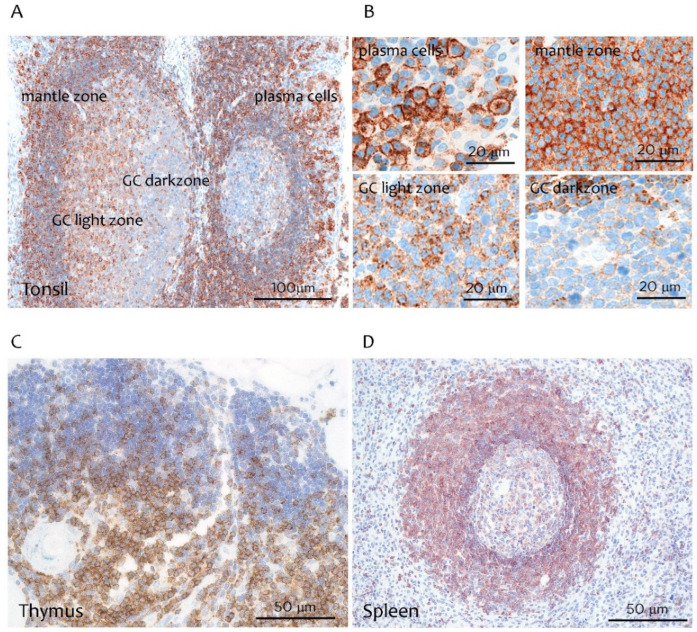
Expression of CD229 on normal lymphoid tissues. (**A**) Single immunoperoxidase labeling with mAb (clone PIZCU426A) against CD229 in human tonsil (magnifications: 20×, scale bar 100 µm). (**B**) Higher magnification of tonsil subepithelial area, mantle zone, GC light zone and GC dark zone (magnifications: 63×, scale bar 20 µm). CD229 staining in Thymus (**C**) and spleen (**D**) (magnifications 40×, scale bar 50 µm).

**Figure 2 cancers-14-02154-f002:**
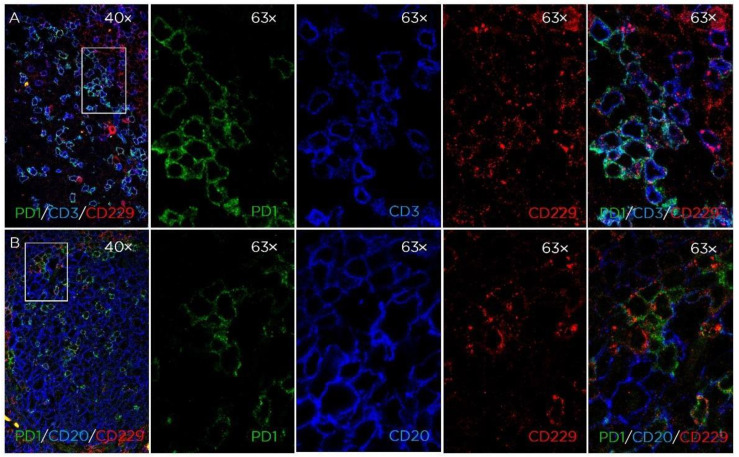
Expression of CD229 in tonsil GC. Triple color immunofluorescence of tonsil GC. (**A**) PD1 (green), CD3 (blue), and CD229 (red); (**B**) PD1 (green), CD20 (blue), and CD229 (red) (magnification: 40× and 63×).

**Figure 3 cancers-14-02154-f003:**
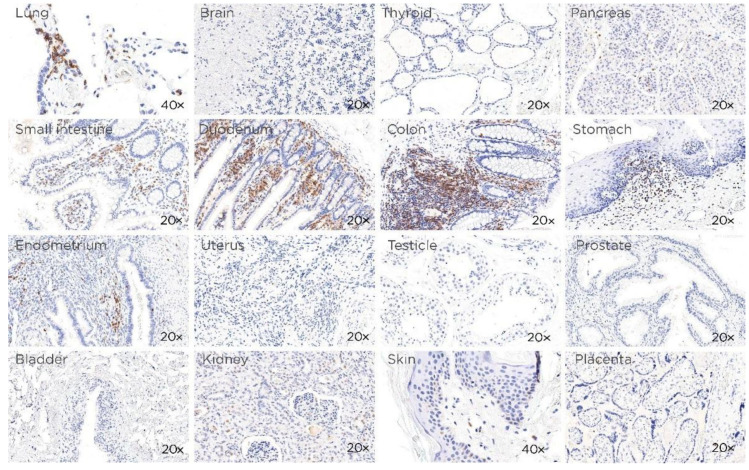
Expression of CD229 in normal tissues. Single immunoperoxidase labeling in reactive human tissues with mAb (clone PIZCU426) against CD229 (magnification: 20× images, brain, thyroid pancreas, small intestine, duodenum, colon, stomach, endometrium, uterus, testicle, prostate, bladder, kidney, skin, placenta and 40× lung and skin).

**Figure 4 cancers-14-02154-f004:**
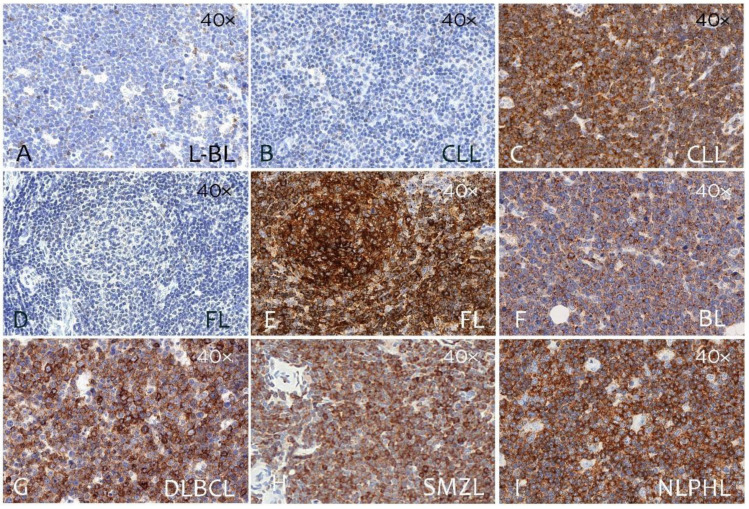
CD229 expression in B-cell lymphomas. Single immunoperoxidase labeling in B-cell lymphomas with mAb (clone PIZCU426) against CD229. Representative examples of B-cell lymphomas (**A**) B-lymphoblastic lymphoma (L-LB), (**B**) negative case, chronic lymphocytic leukemia (CLL) and (**C**) positive case, CLL, (**D**) negative case, follicular lymphoma (FL) and (**E**) positive case, FL, (**F**) Burkitt lymphoma (BL), (**G**) diffuse large B-cell lymphoma (DLBCL), (**H**) splenic marginal-zone lymphoma (SMZL), and (**I**) nodular lymphocyte-predominant Hodgkin lymphoma (NLPHL). All the pictures are at 40× magnification.

**Figure 5 cancers-14-02154-f005:**
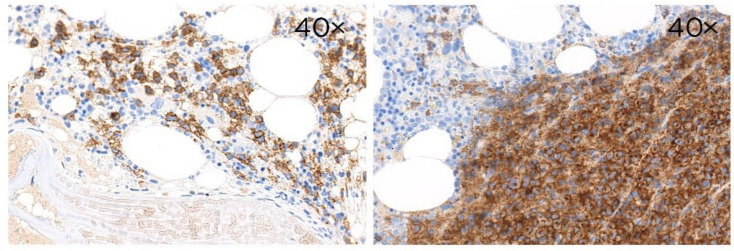
Architectural patterns of bone marrow infiltration by plasma cells by CD229 immunohistochemical stain. Single immunoperoxidase labeling in myeloma with mAb (clone PIZCU426) against CD229. All the pictures are at 40× magnification.

**Figure 6 cancers-14-02154-f006:**
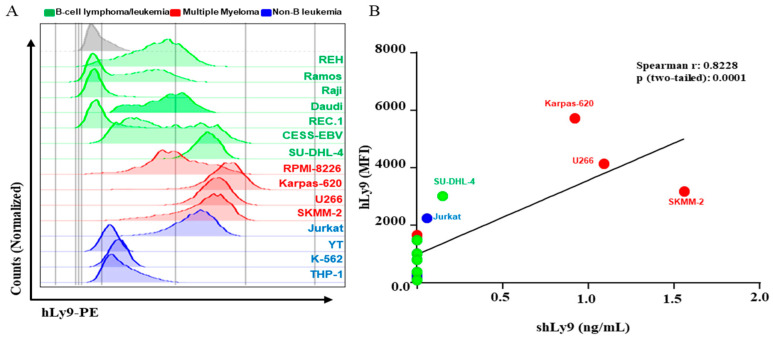
Expression levels of CD229 on cell lines. (**A**) Expression of CD229 on several leukocyte cell lines using flow cytometry. (**B**) Correlation of cell-surface expression (mean fluorescence intensity-MFI) and sCD229 as determined by ELISA.

**Figure 7 cancers-14-02154-f007:**
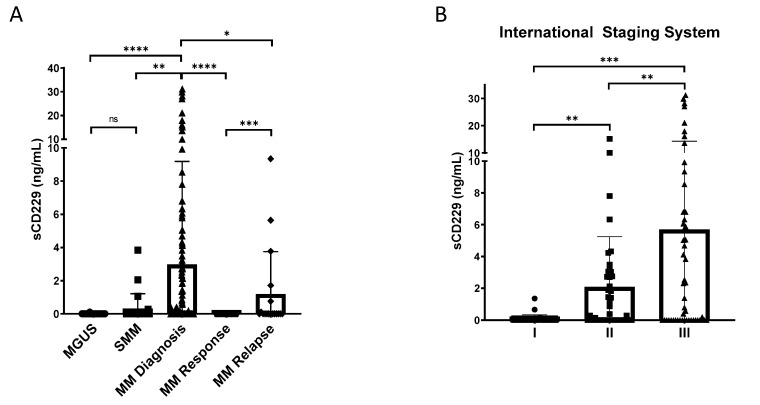
Serum sCD229 levels in patients with MM. (**A**) sCD229 levels in serum samples from 54 patients with MGUS, 23 patients with SMM, 122 patients with newly diagnosed MM, 39 patients achieving response, and 18 patients after myeloma relapse. (**B**) sCD229 levels in serum samples from 122 patients with NDMM according to the International Staging System (ISS). ns, non-significant; * *p* < 0.05; ** *p <* 0.01; *** *p* < 0.001; **** *p* < 0.0001.

**Figure 8 cancers-14-02154-f008:**
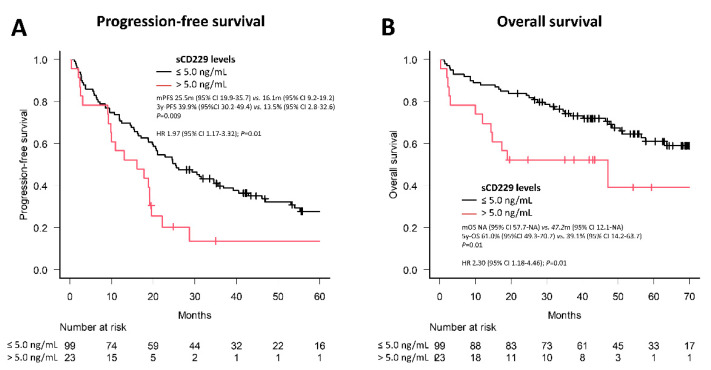
Kaplan-Meier analysis was performed to determine progression-free survival (**A**) and overall survival (**B**) of 122 newly diagnosed MM patients according to whether their baseline sCD229 was above or below the cut-off point of 5.0 ng/mL.

**Table 1 cancers-14-02154-t001:** CD229 expression in lymphomas and myeloma.

Types of Lymphoma	No. Cases	Positive Cases	Negative Cases	% Positive Cases
	Precursor neoplasms			
Precursor B lymphoblastic lymphoma	7	2	5	28%
Mature B-Cell Neoplasms				
Chronic lymphocytic leukemia	20	14	6	70%
Follicular lymphoma	20	12	8	60%
Classic mantle-cell lymphoma	20	14	6	70%
Diffuse large B-cell lymphoma GCB Type	29	19	10	66%
Diffuse large B-cell lymphoma Non-GCB type	19	10	9	52%
Burkitt lymphoma	20	4	16	20%
Nodal marginal-zone lymphoma	34	28	6	82%
MALT lymphoma	28	24	4	86%
Splenic marginal-zone lymphoma	15	13	2	87%
Multiple myeloma ^1^	20	20	0	100%
Hodgkin Lymphomas				
Nodular lymphocyte-predominant HL	7	0	7	0%
Lymphocyte-rich cHL	3	0	3	0%
Nodular sclerosis cHL	10	0	10	0%
Mixed cellularity cHL	8	0	8	0%

^1^ Bone marrow.

**Table 2 cancers-14-02154-t002:** Clinical features of the patients with MM according to the sCD229 cut-off.

Characteristics	sCD229 ≤ 5.0 ng/mL (*n* = 99)	sCD229 > 5.0 ng/mL (*n* = 23)	*p*
Gender, male, *n* (%)	58 (58.6)	11 (47.8)	0.36
Age, median (IQR)	65 (56–73)	69 (66.5–78.5)	**0.01**
Immunological subtype, *n* (%)			0.14
IgG	60 (60.6)	10 (43.5)	
IgA	20 (20.2)	9 (39.1)	
Bence Jones	16 (16.2)	3 (13.0)	
Light chain isotype, *n* (%)			0.09
Kappa	65 (65.7)	10 (43.5)	
Lambda	30 (30.3)	12 (52.2)	
Hemoglobin g/L, median (IQR)	10.8 (10–12.1)	9.6 (8.85–11.4)	**0.01**
Platelets 10^9^/L, median (IQR)	214 (175–171)	177 (129–207)	**0.005**
Calcium mg/dL, median (IQR)	9.3 (8.85–9.8)	9.8 (9.1–10.7)	0.10
Creatinine mg/dL, median (IQR)	0.88 (0.70–1.26)	1.16 (1.04–1.42)	**0.003**
Albumin g/L, median (IQR)	37 (33–42)	32 (28–37)	**0.007**
β2-microglobulin mg/L, median (IQR)	3.85 (2.8–6.1)	8.3 (5.8–11.1)	**<0.001**
Lactate dehydrogensase ≥ ULN, *n* (%)	11 (11.1)	5 (21.7)	0.18
Bone lytic lesions, *n* (%)	69 (70.4)	12 (54.5)	0.35
Bone marrow plasma cells, median (IQR)	28 (13–49.5)	63 (50–73)	**<0.001**
ECOG PS > 2, *n* (%)	15 (15.3)	6 (26.1)	0.23
International Staging System, *n* (%)			**<0.001**
I	32 (32.7)	0 (0)	
II	35 (35.7)	4 (17.4)	
III	31 (31.6)	19 (82.6)	
Cytogenetic abnormalities			0.47
t(4;14)	3 (3.0)	1 (4.3)	
t(11;14)	12 (12.1)	2 (8.7)	
t(14;16)	2 (2.0)	1 (4.3)	
gain(1q+)	3 (3.0)	0 (0)	
del(17p)	0 (0)	0 (0)	
Response, *n* (%)			0.80
Overall response rate	73 (73.7)	16 (69.6)	
Complete remission rate	19 (19.2)	4 (17.4)	1.00
Induction treatment, *n* (%)			0.32
Chemotherapy-based	9 (9.1)	3 (13.6)	
PI-based	32 (32.3)	7 (31.8)	
IMID-based	9 (9.1)	5 (22.7)	
PI + IMI-based	40 (40.4)	5 (22.7)	
Daratumumab-based	6 (6.1)	1 (4.5)	
Elotuzumab-based	3 (3.0)	1 (4.5)	
Autologous stem cell transplantation, *n* (%)	49 (49.5)	3 (13)	**0.002**

ECOG PS, East Cooperative Oncology Group Performance Status; IMID, immunomodulatory drugs; IQR, interquartile range; PI, proteasome inhibitor; and ULN, upper limit of normal. Statistically significant *p*-value in bold.

## Data Availability

The data presented in this study are available on request from the corresponding author.

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
