# Peer review of "CD229 (Ly9) a Novel Biomarker for B-Cell Malignancies and Multiple Myeloma"

_cancers, 2022, doi:10.3390/cancers14092154_

Round 1

Reviewer 1 Report

Thank you for re-considering this manuscript and thanks to the author for addressing my previous comments. The main concern regarding the first draft was that the authors highlighted their core finding of the manuscript as CD229 expression on B cell malignancies. A finding that was lacking the required novelty for publishing the article. The authors have worked on their findings and highlighted them from a different perspective that assured novel results. Now, they show that CD229 expression is restricted to lympho-hematopoietic tissues and mainly on B cell linage, moreover they show for the first time that CD229 has higher expression levels on mature B cells. So one can conclude that first concern was addressed.

The second main concern was the clinical findings and the lack of multivariable analysis. I required back then that a prober stratification analysis should be performed during the analyses as I found no clear proof that CD229 can cause bad progression in newly diagnosed patients with multiple myeloma. After this revision, the authors have strongly addressed this problem. In their clinical analysis, different variables have been included and tested with regard to worse prognosis in patients. Multivariate Cox regression models were used to test such variables in Progression Free Survival and Overall Survival analysis. All in all, the authors have addressed the missing points and that these findings are of interest to the broad and translationally interested readership of your esteemed Journal."

Author Response

We want to thank the reviewer for finding that our revised version has addressed his/her comments and finds our paper of interest to the readership of CANCERS.

Reviewer 2 Report

The reviewed version of this manuscript addresses most of the questions and comments made.

My understanding is that one of the strengths of this work is the development of an anti CD229 antibody to work on FFPE tissue and the description of the distribution in normal tissue and lymphoid malignancies. It is still unclear how was the 10% cut off for positivity in immunohistochemistry determined.

Minor points should be corrected in the text. For instance, the title of Table 1 should be "CD229 expression in lymphomas and myeloma", and the specimens used (lymph node, bone marrow) should be mentioned as footnotes. 

On line 507 "...studying myeloma cells in blood samples" should be corrected (the studies were made in bone marrow cells) 

The manuscript is clearer now but it still may benefit from spell checking and minor language corrections.

Author Response

We thank the review for his/her insightful comments that have help us to improve our manuscript.

We have added at statement in M&M to clarify how the 10 cut off of positivity in the immunohistochemistry was determined.

These is the new paragraph:

CD229 protein expression was assessed by two independent observers (J.F.G and MRJ) by immunohistochemistry on reactive as well as neoplastic human TMA and on complete sections. Since most tumors showed uniform immunohistochemical expression, we selected  a score of 10% as the most informative cut off, and each case was scored semi-quantitatively, depending on the number of positive cells, as negative (0–10% positive tumor cells) or positive (10–100% positive tumor cells)

Following her/his advice with have corrected the text (eee attached edited version of our manuscript.)

This manuscript is a resubmission of an earlier submission. The following is a list of the peer review reports and author responses from that submission.

Round 1

Reviewer 1 Report

This is a clearly written paper focusing on the characterization of the expression of CD229 on normal tissues and B cell malignancies with a new antibody suitable for immunohistochemistry, and on the role of soluble CD229 as a prognostic marker in multiple myeloma. Although the expression and role of CD229 had been already explored in some studies in multiple myeloma, this was not the case for B cell lymphomas. The design of the study is adequate. Nevertheless the prognostic role of sCD229 in multiple myeloma is difficult to interpret without multivariate analysis for prognostic factors, since the population with highest values had multiple well known poor prognostic characteristics.

Specific issues:

In the “Simple Summary” and “Abstract” MALT lymphomas are mentioned as expressing CD229, however none is studied in the paper – this should be deleted or explained.

-In the “Methods” section the criteria/cut off (and method of determination) for CD229 positivity in the immunohistochemistry studies is not explained. This makes the interpretation of part of the results difficult, namely the ones in B cell lymphomas.

-In point 2.5 the expressions “many of the normal and tumor samples were included in several…” are imprecise.

-Was the analysis of stained slides and TMAs automatically performed?

-In the statistical methods no multivariate analysis was described. The absence of multivariable analysis may turn difficult the interpretation of the prognostic impact of sCD229 levels in multiple myeloma cases.

-In the “Results” section, point 3.2, the staining for CD229 (positive or negative) should be explained in relation to a cut off.

-Statements related to marginal zone lymphoma concern only seven cases of splenic MZL, making it difficult to generalize the conclusions. Were any MALT or nodal MZL (or lymphoplasmacytic lymphomas) studied?

-The interpretation of only 35% positivity in diffuse large B cell lymphomas, in contrast with 60% positivity in follicular lymphoma, deserves a comment in the Discussion.

-Section 3.4 explores the prognostic role of sCD229. The definition of prognosis (impact on PFS and/or OS) should be included in the text. PFS should be defined – after first line treatment?

-The experimental results are clear but it is difficult to evaluate the independent role of the sCD229 marker without a multivariate analysis, since it segregates with well-known poor prognosis factors.

-Also, the treatment modalities used are not described, and this is relevant for the evaluation of prognosis

In the Discussion it would be interesting if the authors could elaborate on the role of CD229 as a therapeutic target, namely after pointing out the possible problems with the potential “on target off tissue” toxicities of cell therapies.

Minor comment

In page 9, line 315, I believe there is a mistake: where CD22 is written there should be CD229

Reviewer 2 Report

The article “CD229 (Ly9) a novel biomarker for B cell malignancies and multiple myeloma” by Roncador G et al. reported the generation of a new anti-CD229 (Ly3, SLAMF3) monoclonal antibody clone PIZCU426A and their utility in detecting B cell malignancies, especially mature B cell malignancies. While authors showed the favorable potency of PIZCU426A in detecting CD229 expression, the high expression of CD229 in MM and mature B cell has been already well known, and CD229 has been already focused as the therapeutic target in future immune therapy. Unfortunately, this article provided no significant novel findings of CD229 in the disease biology, therapy or diagnostics in cancers, especially in lymphoid malignancies, as this article just showed the expression patterns of CD229 in B cell malignancies. Overall, the article is out of scope of the journal Cancers, but may be more suitable for journals for histopathology, diagnostics, or laboratory medicine. In addition, there are several concerns with this article.

Comment

  1. First, the major topic of this article is the generation of a new antibody PIZCU426A. While authors described the lack of anti-CD229 Ab before this new antibody, we can easily find several commercially available antibodies for CD229 which are reported to be available for immunohistochemistry and flow cytometry on website etc. What are the major or specific advantages of PIZCU426A compared with other previous antibodies, and what are the problems with previous antibodies? These points were totally unclear with this article.
  2. As authors discussed, the association with CD229 expression pattern and disease subtype is critically important in CLL (naïve vs. memory), MCL (histologic subtypes line indolent, conventional, and blastoid) and DLBCL (classification of cell of origin, etc). These are expected to be addressed in this study, so that the article may increase the scientific impact by adding novel findings.
  3. In Figure 4, more precise explanation is needed. What do Figure 4B and Figure 4D mean? Are they the representatives of negative results?
  4. In association with Figure 6B, data about the relationship between CD229 expression and serum sCD229 in clinical samples are of interest.
  5. Considering the association between disease progression and the elevation of sCD229, the association between CD229 expression and disease progression is expected to be addressed.
  6. Authors defined the cut-off level of sCD229 to be 5.0 ng/ml in this article just by examining sCD229 levels in sera from patients with B cell malignancies. What about the normal range of serum sCD229 in healthy individuals? At the moment, we are not sure whether serum sCD229 level is higher in patients with B cell malignancies compared with healthy controls.
  7. Prognostic value of sCD229 needs to be evaluated by the multivariate analysis with factors of ISS.
  8. Considering the gene location, it would be also important to address the association between high CD229 expression and chromosome 1q gain in multiple myeloma. Also, it it expected to discuss the underlying mechanism for CD229 overexpression in mature B cell tumors.
  9. On page 5, line 244. The description “August Pi I Sunyer Biomedical Research es and tonsil” does not male sense.
  10. On page 9, line 314. The term “CD22” might be “CD229”.

Reviewer 3 Report

Thank you for the opportunity to review this manuscript. In this work by Roncador and her colleagues, the research team is providing new insight into the expression of the SLAM family immunoreceptor CD229 in different tissues. SLAM family receptors are type I transmembrane glycoproteins that belong to the immunoglobulin (Ig) superfamily. They are a group of adhesion molecules widely expressed on different hematopoietic cells and are involved in many adaptive immunity functions. The main finding of the manuscript is the unique expression of CD229 expression in B cell malignancies, the authors used a monoclonal antibody against the cytoplasmic region of CD229 to identify its expression. Moreover, the authors used ELISA quantification approach to show that serum level of CD229 in myeloma patients is a prognostic marker.

Although the manuscript provides additional evidence to the already existing knowledge of CD229 expression, it lacks the requisite novelty for an original article. Moreover, the manuscript suffers major flaws in the experimental/statistical setups that hinder its acceptance:

  • The core finding of the manuscript that CD229 is expressed in B cell malignancies has been described and validated previously. On 2011 Atanackovic and his colleagues have identified the marker and described it as a useful diagnostic marker for Myeloma patients: https://haematologica.org/article/view/6099
  • The expression of CD229 on B cell malignancies has been validated for years and have led to the development of anti-CD229 CAR T cells by Radhakrishnan and their colleagues: https://www.nature.com/articles/s41467-020-14619-z
  • The second main finding was that the median progression-free survival (PFS) was shorter for Newly diagnosed Myeloma patients with sCD229 levels >5.0 ng/ml compared to those with levels ≤5.0 ng/ml. A prober stratification analysis was missing in the analyses. The authors even highlighted the fact that in the “sCD229 levels >5.0 ng/ml” group, patients were significantly older, had a lower level of hemoglobin, and a very high β2-microglobuline, a finding that excludes any prognostic value for CD229.  
  • A key multivariable analysis is also missing, authors had to report estimated effects with confidence intervals for the CD229 and all other variables in the model.
  • The authors also did not provide any information about the therapy the patients received during the trial.